# Controlled Fermentation Using Autochthonous *Lactobacillus plantarum* Improves Antimicrobial Potential of Chinese Chives against Poultry Pathogens

**DOI:** 10.3390/antibiotics9070386

**Published:** 2020-07-07

**Authors:** Damini Kothari, Woo-Do Lee, Eun Sung Jung, Kai-Min Niu, Choong Hwan Lee, Soo-Ki Kim

**Affiliations:** 1Department of Animal Science and Technology, Konkuk University, 120 Neungdong-ro, Gwangjin-gu, Seoul 05029, Korea; damini.kth@gmail.com (D.K.); caw147@naver.com (W.-D.L.); 2Department of Bioscience and Biotechnology, Konkuk University, 120 Neungdong-ro, Gwangjin-gu, Seoul 05029, Korea; jes708@naver.com; 3Institute of Biological Resource, Jiangxi Academy of Sciences, Nanchang 330029, China; niukaimin@naver.com

**Keywords:** antibiotic alternative, Chinese chives, fermentation, *Lactobacillus plantarum*, poultry feed additive

## Abstract

Chinese chives (CC) are rich in several antimicrobial constituents including organosulfur compounds, phenolics, and saponins, among others. Herein, we fermented CC juice using an autochthonous isolate, *Lactobacillus plantarum* having antimicrobial effects against poultry pathogens toward formulating an antimicrobial feed additive. Following 24 h of fermentation, the antimicrobial and antiviral activities of CC juice were significantly enhanced against poultry pathogens. However, the antioxidant activity of CC juice was significantly decreased following fermentation. Meanwhile, the compositional changes of CC juice following fermentation were also investigated. The total polyphenol, thiol, and allicin contents were significantly decreased in *L. plantarum* 24 h-fermented CC juice (LpCC) extract; however, total flavonoids increased significantly following fermentation. The untargeted metabolite profiling of nonfermented CC juice (NCC) and LpCC extracts was carried out using the ultra-high-performance liquid chromatography-linear trap quadrupole-orbitrap-tandem mass spectrometry (UHPLC-LTQ-Orbitrap-MS/MS) followed by multivariate analyses. The score plots of principal component analysis (PCA) and orthogonal partial least squares-discriminant analysis (OPLS-DA) based on UHPLC-LTQ-Orbitrap-MS/MS datasets displayed a clear segregation between the LpCC and NCC samples, which suggests their marked metabolomic disparity. Based on the multivariate analysis, we selected 17 significantly discriminant metabolites belonging to the different chemical classes including alkaloid, flavonols, saponins, fatty acids, amino acids, and organic acids. Notably, the flavonols including the glycosides of quercetin, kaempferol, and isorhamnetin as well as the saponins displayed significantly higher relative abundance in LpCC as compared with NCC. This study provides useful insights for the development of a fermented CC juice based antimicrobial feed additive to combat poultry infections.

## 1. Introduction

Poultry is one of the most widespread food industries worldwide and is expanding continuously at an unprecedented rate. The burgeoning poultry demand is often achieved by the rampant use of antibiotics which can lead to the emergence of resistant pathogens [1,2]. The prevalence of antibiotic resistant pathogens threatens animal and human health coupled with the colossal economic losses worldwide due to treatment failure [3]. In Europe, more than 1500 deaths are directly related to antibiotic use in poultry during 2007 [4]. Nevertheless, it is very disconcerting that antibiotic usage in poultry sector is increasing every year and without any strict regulatory legislation, it is expected to grow by 143% over the next 10 years globally [5]. Several promising antibiotic alternative approaches such as phytogenic additives, prebiotics, probiotics, synbiotics, organic acids, antimicrobial peptides, bacteriophages, antibodies, and feed enzymes have been reported [6]. However, these alternatives are not individually competent enough to meet the efficiency goals of antibiotics in poultry. Therefore, the development of an effective as well as broad-spectrum antibiotic alternative without impairing the production is a thrust area of poultry research.

In the present study, we considered a combinational approach including “phytogenic and probiotic” entities to develop an effective antibiotic alternative. The plant genus *Allium* (for example, garlic and onion) exhibits promising antimicrobial activities owing to the high proportions of organosulfur compounds, phenolics, and saponins, among others [7,8,9,10]. Recently, garlic was shown to improve poultry feed as an effective antimicrobial agent with targeted toxicity towards pathogens, especially against multi-drug-resistant bacteria like *Salmonella* Typhimurium [7] and *Escherichia coli* O78 [11]. Among the several *Allium* species, we examined *A. tuberosum* commonly known as ‘Chinese chives’ (CC), an all-season abundant herb used in Asian cuisines as potential source of poultry feed additive.

For thousands of years, lactic acid bacteria (LAB) mediated fermentation has been used to preserve and improve foods. Several studies have reported that *Allium* fermentation has enhanced biofunctionalities through altering the composition of bioactive components including polyphenols, free amino acids, organosulfur compounds, vitamins, and dietary fibers [8,10,12,13]. The fermentative changes in the composition of plant matrices are strain- and species-specific [14]. Most of the previous studies on *Allium* fermentation were focused on either spontaneous fermentation [13] or heat-treated plant matrices [12]. However, spontaneous fermentation may not be an authentic system to obtain a safe and palatable final product as the possibility of detrimental microbiological/toxicological consequences such as biogenic amines, mycotoxins, and pathogenic microbes, could not be excluded [15]. Recently, Lee et al. [16] indicated the involvement of autochthonous LAB in biogenic amine accumulation during spontaneous fermentation of green onion kimchi. Although thermal treatments are effective in the elimination of autochthonous microorganisms and endogenous enzymes from plant matrices, they irreversibly deteriorate the associated biofunctionalities through altering the chemical composition [17,18]. Reportedly, the antimicrobial activities of some *Allium* spp. have shown to be decreased following the heat treatments [9]. Hence, we argue that microbial fermentation of a filter-sterilized plant matrix with characterized inocula would be more suitable toward the biotransformation of plant-derived compounds without compromising the basic chemical compositions and associated biofunctionalities.

To the best of our knowledge, no reports have been published so far which describe the fermentative bioprocess for CC juice toward the potential development of a poultry feed additive. Considering the valuable properties of CC as a relatively cheap and rich source of natural antimicrobials compounds, we examined the effects of fermentative bioprocess using an autochthonous isolate on the chemical composition of CC juice toward poultry feed additive development. Herein, *L. plantarum* isolated from CC juice was used for the controlled fermentation of filter-sterilized CC juice. Further, liquid chromatography–tandem mass spectrometry (LC-MS/MS) based untargeted metabolomic profiling was employed to examine the discriminant metabolites following the fermentative bioprocess.

## 2. Results

### 2.1. Selection of Autochthonous Starter Culture

In the spontaneously fermented samples with deMan, Rogosa and Sharpe (MRS) broth, we isolated and identified *Lactobacillus sakei*, *L. plantarum*, *Leuconostoc mesenteroids*, *Weissella cibaria*, and *W. paramesenteroides* (Table 1). However, in Luria-Bertani (LB) broth, *Bacillus megaterium*, *B. aryabhattai*, *B. pumilus*, and *B. subtilis* were identified in addition to some pathogenic strains including *Staphylococcus sciuri* and *Micrococcus luteus*. In yeast malt (YM) broth, only *Saccharomyces cerevisiae* was isolated and identified. Among all the isolates, only *L. plantarum* displayed notable antibacterial effects against the tested poultry pathogens and was hence selected as the starter inoculum toward axenic fermentation of CC juice in subsequent experiments (Table 1). The screened isolate, i.e., *L. plantarum*, was submitted to the Korean Agricultural Culture Collection (KACC) and designated as KACC 92270P.

### 2.2. Microbial Population and pH Changes During Fermentation of CC Juice

The microbial growth and pH of the CC juice during the 24 h fermentation are shown in Figure 1. The viable cell counts of *L. plantarum* reached at 8.11 log_10_ colony forming units (CFU)/mL within 6 h of fermentation, reached a maximum of 8.8 log_10_CFU/mL at 18 h and then maintained at the same level until 24 h. Notably, the pH of the CC juice containing MRS broth rapidly decreased to 4.9 from an initial value of 6.1 within 12 h of fermentation and further dropped to 3.95 after 24 h of fermentation.

### 2.3. Changes in the Bioactivities of CC Juice Following the L. plantarum Mediated Fermentation

#### 2.3.1. Antibacterial Activity

The antibacterial activity of non-fermented CC juice (NCC) and 24 h-*L. plantarum* fermented CC juice (LpCC) supernatants against eleven common poultry pathogens is shown in Figure 2A. Notably, a positive antimicrobial activity of both fermented and non-fermented CC juice samples against all the tested pathogens was observed. Intriguingly, the antibacterial activity of LpCC was significantly higher against *C. perfringens* (*p* < 0.01), *E. faecalis* (*p* < 0.01), *S.* Anatum (*p* < 0.01), *S.* Enteriditis (*p* < 0.05), *S.* Gallinarum (*p* < 0.01), *S.* Pullorum (*p* < 0.01), *S.* Paratyphi (*p* < 0.01), *S.* Typhi (*p* < 0.01), *S.* Typhimurium (*p* < 0.01), and *St. aureus* (*p* < 0.01) as compared with NCC. In addition, the zones of inhibition with LpCC exhibited no change over time. However, it tended to disappear in the case of NCC (data not shown).

#### 2.3.2. Antiviral Activity

In the present study, an *in ovo* model was employed for studying the antiviral activities of LpCC and NCC extracts against a low-pathogenic avian influenza virus, H1N1. The hemagglutination-based assay displayed absence of virus titer in chick embryos, indicating the inhibitory potential of CC juice extracts (Figure 2B). LpCC displayed significant antiviral activity at a lower dose rate of 10 mg/mL when compared with NCC at the same concentration (*p* < 0.001). However, lower doses of these extracts failed to elicit any antiviral activity in chick embryos. In addition, NCC and LpCC did not induce any cytotoxicity up to 50 mg/mL i.e., 1 to 10 mg/chick embryo (data not shown).

#### 2.3.3. Antioxidant Activity

The antioxidant activity of CC juice extracts (NCC and LpCC) was evaluated using ABTS and DPPH radical scavenging assays. As shown in Figure 2C, the 2,2’-azino-bis(3-ethylbenzothiazoline-6-sulfonic acid (ABTS) and 2,2-diphenyl-1-picrylhydrazyl (DPPH) radical scavenging activities of LpCC were significantly decreased by 37% and 28%, respectively, when compared to NCC (*p* < 0.001).

### 2.4. Biochemical Changes

The results of biochemical constituents in the methanol extracts of CC juice before and after fermentation are shown in Figure 3. The total polyphenols (*p* < 0.01) and organosulfur compounds, including thiols (*p* < 0.05) and allicin (*p* < 0.001) were significantly decreased after a 24 h fermentation of CC juice (−13%, −5%, and −21%, respectively, compared to NCC). However, the total flavonoid content of CC juice was significantly increased by 6% following fermentation (*p* < 0.05).

To analyze the effect of *L. plantarum* mediated fermentation on CC metabolites, an untargeted metabolite profiling of LpCC and NCC extracts using ultra-high-performance liquid chromatography-linear trap quadrupole-orbitrap-tandem mass spectrometry (UHPLC-LTQ-Orbitrap-MS/MS) was carried out. As shown in Figure 4A, the principal component analysis (PCA) score plot displayed a clearly segregated pattern between LpCC and NCC metabolomic datasets across PC1 (37.3%) and PC2 (10%). Similar clustered patterns were evident between the datasets in supervised orthogonal partial least squares-discriminant analysis (OPLS-DA) model with an overall variance of 35% coupled with high predictive ability (Q2 = 0.971), high predictive variations (R2X = 0.468 and R2Y = 0.995), as well as considerable significance metric (*p* = 7.02 × 10^−10^), as shown in Figure 4B. Overall, we tentatively identified 20 secondary metabolites falling under different chemical classes including six flavonoid family metabolites, five saponins (structure unidentified), three fatty acids derived compounds, two amino acids, one alkaloid, two organic acids, and one benzofuran coumarin (Table 2).

We selected the significantly discriminant metabolites adding to the observed variance between the LpCC and NCC extracts using the OPLS-DA model at a variable importance for projection (VIP) > 0.7 and *p* < 0.05. A total of 17 metabolites were considered significantly discriminant with clearly distinct relative abundance between LpCC and NCC extracts (Figure 5A–Q). In NCC, the relative contents of several metabolites including N-(1-deoxy-1-fructosyl) phenylalanine **(1)**, glycyrol **(3)**, tryptophan **(4)**, and tianshic acid **(19)** were significantly higher than those in the LpCC extract (Figure 5A,C,D,P). Conversely, the LpCC extract displayed significantly increased levels of 3-(2,3,4-trihydroxy-5-methoxyphenyl) propanoic acid **(2)**, benzoylmesaconine derivative **(5)**, kaempferol-diglucoside **(7)**, isorhamnetin 3,4′-diglucoside **(8)**, quercetin-diglucoside **(9)**, quercetin-hexoside **(10)**, kaempferol-glucoside **(14)**, 12-hydroxystearic acid **(20)**, and five unidentified saponins **(12, 13, 15, 16, 17)** (Figure 5B,E–O,Q).

## 3. Discussion

Selection of autochthonous starters represents a useful tool to develop fermented products with improved quality and safety [19]. This study aimed at isolating and selecting autochthonous bacteria from CC juice having antimicrobial activity against poultry pathogens, and to examine its suitability as a fermentation starter toward developing an antimicrobial feed additive. In general, LAB constitute a small part (2.0–4.0 log_10_ CFU/g) of the autochthonous microbiota of raw vegetables and fruits. Most notably, the genera including *Leuconostoc*, *Lactobacillus*, *Weissella*, *Enterococcus*, and *Pediococcus* are frequently identified from various vegetables [20]. In the present study, an autochthonous strain *L. plantarum* was isolated and selected based on its broad range of antimicrobial activities against the tested poultry pathogens. In previous years, several studies have reported the antimicrobial activity of *L. plantarum* against poultry pathogens under both in vitro and in vivo conditions. Notably, Dec et al. [21] had reported that *L. plantarum* of domestic goose origin exhibited strong antagonism toward several poultry pathogens *in vitro*. Recently, Wang et al. [22] reported both in vitro and in vivo antibacterial activities for *L. plantarum* of chicken origin. The ability of a bacterial strain to restrict the growth of pathogens has been proposed as one of the criteria for the selection of probiotics in many studies. Moreover, antimicrobial activity of *L. plantarum* is thought to be an important mean to competitively exclude or inhibit invading spoilage bacteria in a fermented product. The pH value is another factor which influences the extent of fermentation. In the present study, fermentation of CC juice by autochthonous *L. plantarum* resulted in subsequent pH drop (ca. 3.95) at 24 h concomitant to the exponential growth phase (ca. 9 log_10_ CFU/mL), which may prevent proliferation of spoilage organisms and thereby maintaining the quality characteristics related to yeast outgrowth during the storage period [23,24]. Our results were in accordance with the previous reports which also indicated that *L. plantarum* plays a dominant role in the autochthonous fermentation of vegetables [25].

The *Allium* spp. display antibacterial activities against several Gram-positive and Gram-negative bacteria owing to their high contents of organosulfur compounds [26,27]. However, the antimicrobial activities decline quickly and are even lost pertaining to the high volatility and instability of these compounds [13]. The stronger antibacterial activity of LpCC might be attributed to lactic acid and short-chain organic acids, released following the fermentation. In addition, the antimicrobial activities of the fermented CC juice might also be attributed to the specific compositional changes in CC juice following the *L. plantarum* mediated fermentation.

In general, *L. plantarum* is widely considered as the common starter culture in food fermentation which can enhance the fermentative bioprocess coupled with the release of desired metabolites maneuvering the organoleptic characteristics of end products [28,29]. In this study, *L. plantarum* mediated fermentation of CC juice was characterized with the decrease in total polyphenols which may be ascribed to their direct utilization by microorganisms while the fermentative growth. The loss of polyphenols following fermentation resulted in the reduced antioxidant activity in CC juice (LpCC). Previously, Othman et al. [30] also reported the loss of phenolic compounds during fermentation from olives. However, significantly increased total flavonoid content (TFC) in LpCC may be attributed to the release of bound phenolics following fermentation [31]. Nevertheless, the TPC and TFC could not always serve as proxies for biological activities of a plant matrix, instead key metabolites underlying a process or interaction might be better indicators. Generally, thiols and allicin are responsible for the typical strong odor of *Allium* spp. Herein, the decrease of thiol and allicin contents of CC juice following fermentation may be attributed to their volatile nature [32] as well as the simultaneous utilization by the lactobacilli during fermentation [33]. The reduced organosulfur contents found in our study are comparable to the findings reported by Yang et al. [34] Another study showed that garlic fermentation resulted in a significantly lower concentration of γ-glutamyl peptide [12], a key player in flavor precursor biosynthesis [35]. Indeed, in an in vivo study, an improved feed intake with the fermented garlic supplementation in broilers was observed [36]. In line of this consideration, we conjecture that fermentation may improve the flavor of CC juice by reducing the strong odor related to thiols and allicin.

Metabolomics is a promising approach to track the compositional changes in the plant matrices and screen potential biomarkers for the mechanistic understanding of fermentative bioprocess concerning plant–microbe interactions. Herein, the multivariate statistical analysis of UHPLC-LTQ-Orbitrap-MS/MS based untargeted metabolomic data sets unraveled the substantial disparity between the secondary metabolite profiles of NCC and LpCC extracts, signifying the effects of *L. plantarum* mediated fermentation (Figure 4). Flavonols including quercetin, kaempferol, myricetin, rutin, and isorhamnetin, as well as their derivatives are among the most important class of *Allium* flavonoids with potent antibacterial [37] and antiviral [38] activities. Recently, Al-Yousef et al. [39] reported that onion peel extracts rich in ‘quercetin 4-O-β-D glucopyranoside’ have potential anti-infective properties owing to its anti-QS and anti-biofilm properties against pathogens. A kaempferol-3-rutinoside isolated from *Sophora japonica* flowers exhibited a potent inhibitory activity against *Streptococcus mutans* derived ‘sortase A’ that plays a key role in the adhesion and subsequent host invasion by Gram-positive bacteria [40]. The antiviral activity of CC juice extracts (NCC and LpCC) *in ovo* was demonstrated in the current study by a diminution of the allantoic fluid hemagglutination ability, as an indicator of the number of viruses present in the live chicken embryos. Previously, Ghoke et al. [41] indicated that plant polyphenols possess high binding affinities with the two virion targets, namely hemagglutinin and neuraminidase on the surface of virus, which might be a reason for their antiviral activity. Surprisingly, we found a potent antiviral activity with the lower concentration of LpCC, i.e., ≥10 mg/mL as compared with NCC i.e., ≥25 mg/mL. The CC juice displayed raised levels of typical flavonoids following the autochthonous fermentation, which may be an explanation for heightened antagonistic activities in LpCC. In addition, the elevated levels of oxo-dihydroxy-octadecenoic acid (plant oxygenated fatty acid) (VIP < 0.7), 12-hydroxystearic acid, and saponins following fermentation might also have contributed to the enhanced antagonistic activity of the fermented juice (Figure 5). These results indicate that the improved antagonistic activity of fermented CC juice against poultry pathogens is a complex phenomenon, which might involve synergistic actions of the multiple coexisting metabolites present in the extracts. Although the untargeted metabolomics provides comprehensive insights into complex metabolomes of biological samples, it is often limited by the incomplete availability of spectral references, large-scale, and unambiguous annotation of unknown compounds [42]. Therefore, future targeted studies with authentic standards are warranted toward further validation of these results.

## 4. Materials and Methods

### 4.1. Isolation, Identification, and Screening of Microorganisms from CC Juice

CC were purchased from a local market (Seoul, Republic of Korea) and juice was prepared using a juicer (Angel-juicer, Busan, Republic of Korea). To isolate indigenous microorganisms from CC, 1 mL of CC juice was mixed with 9 mL of MRS, YM, and LB broth. The mixtures were subjected to spontaneous fermentation at 30 °C and 100 rpm for three days. Following the incubation, spread plating was done using appropriate dilutions on the respective media and incubated at 30 °C for 48 h. The morphologically distinct colonies were picked, streaked again until purification, and identified using 16S rDNA sequence analysis by a commercial service, Macrogen Inc. (Seoul, Republic of Korea). The similarity of 16S rDNA sequences were analyzed using GenBank Basic Local Alignment Search Tool (BLAST). A total of 12 isolates were identified and stored in their respective media and glycerol (15%, *v/v*) at −80 °C. The identified microorganisms were screened based on their antimicrobial activities against 11 poultry pathogens using the agar well diffusion method. The poultry pathogens, including different *Salmonella enterica serovars* such as *S.* Gallinarum, *S.* Enteritidis, *S.* Pullorum, *S.* Typhimurium, *S.* Anatum, *S.* Typhi, and *S.* Paratyphi, *C. perfringens*, *St. aureus*, *E. coli*, and *E. faecalis* were obtained from Korean National Veterinary Research and Quarantine Service (NVRQS, Republic of Korea). The overnight grown pathogen cultures were swabbed on nutrient agar (NA) plates and 100 µL of cell-free culture supernatant (CFCS) of all the isolates filtered with 0.22 µm membrane filters was added into the wells (6 mm diameter) on each agar plate. All the plates were incubated at 37 °C for 18 h and the clear zone of inhibition (mm) was measured.

### 4.2. Fermentation of CC Juice

To carry out controlled fermentation, CC juice was centrifuged at 23,140× *g* for 10 min and collected supernatant was filter sterilized using 0.2 µm membrane bottle-top filters (Nalgene, USA). The filter-sterilized juice was added into MRS broth at a concentration of 25% (*v/v*) and then inoculated with ca. 10^7^ log_10_ CFU/mL of *L. plantarum* without adjusting the pH (pH ~ 6). A similar mixture without inoculum was used as a negative control (NCC). The fermentation was carried out in triplicate at 30 °C and monitored at 6, 12, 18, and 24 h by measuring the viable cell count and pH. Enumeration of viable bacteria was carried out by plating onto MRS agar at 30 °C for 48 h.

### 4.3. Antibacterial Activity Using Agar Well-Diffusion Assay

The NCC and LpCC samples were centrifuged at 10,000× *g* for 5 min at 4 °C. Antibacterial activities of the membrane filtered (0.22 µm) cell-free supernatants of LpCC and NCC were comparatively examined using agar well diffusion method against 11 aforementioned poultry pathogens as described in the Section 4.1.

### 4.4. Preparation of Extracts

The freeze-dried supernatants (5 g) of NCC and LpCC samples were extracted using 50 mL of 80% methanol. First, the samples were sonicated for 30 min and kept under deep-freezing (−20 °C) conditions overnight toward achieving better precipitation of high molecular weight compounds. The next day, each of the mixture was centrifuged at 23,140× *g* for 30 min and the supernatant was vacuum dried at 40 °C, lyophilized and stored at −20 °C until analyses.

### 4.5. Antiviral Activity Using Hemagglutination Assay

The antiviral efficacies of NCC and LpCC extracts in the specific pathogen-free (SPF) embryonated chicken eggs were estimated using hemagglutination (HA) assay. Briefly, a varying sample concentration ranging from 5 to 50 mg/mL were subjected to incubation with low-pathogenic influenza virus (1:1) at 4 °C for 30 min. Aliquots (0.2 mL) of the mixture of CC juice samples, or PBS and H1N1 (10^6^ EID_50_/mL, 50% egg infective dose per mL) were injected into allantoic cavity of 10-day-old SPF embryonated eggs (five for each treatment) separately. The inoculated site was sealed with paraffin wax and the inoculated eggs were incubated for five days at 35 °C with the air sac uppermost. Following incubation, the oocyte lysates (0.2 mL) were collected and hemagglutination assay was performed using 1% (*v/v*) chicken red blood cells at room temperature. The virus titers in the assay were expressed as the EID_50_/mL.

### 4.6. Antioxidant Assays

The antioxidant activity of NCC and LpCC extracts was determined using ABTS [43] and DPPH [44] radical scavenging assays with some modifications. For the ABTS assay, a working solution was prepared by mixing equal volumes of 7 mM ABTS and 2.45 mM potassium persulfate solutions and incubated for 16 h at room temperature in the dark. The resulting solution was diluted with water until the absorbance value reached ~ 0.7 ± 0.02 at 734 nm. An antioxidant assay was performed by adding 180 µL of ABTS working solution to 20 µL of each CC juice extracts (5 mg/mL) in a 96-well plate, and the reaction was incubated for 6 min at 37 °C in the dark. The resulting sample absorbance was recorded at 734 nm using an automated microplate reader (BioTek, USA) and 80% (*v/v*) methanol was used as a control.

In the DPPH radical scavenging assay, 20 µL of sample extracts (5 mg/mL) were added into 180 µL of 0.2 mM DPPH solution in 80% (*v/v*) methanol in a 96-well plate. The mixtures were incubated in the dark at room temperature for 30 min and reaction absorbance was read at 517 nm in an automated microplate reader using methanol (80%, *v/v*) as a control. The scavenging activity of the ABTS and DPPH radicals was calculated by the equation (1).
(1)$$\mathrm{Radical}\;\mathrm{scavanging}\;\mathrm{capacity}\;\left(\%\right)=1-\left(\frac{A_{s}}{A_{c}}\right)\times 100$$
where, A_s_ and A_c_ are the absorbance of samples and control, respectively.

### 4.7. Biochemical Constituent Analysis

#### 4.7.1. Total Phenolic Content (TPC)

TPC of the NCC and LpCC extracts (10 mg/mL) was comparatively assayed following the protocol adapted from Suh et al. [43]. In a 96-well microtiter plate, 20 µL of the samples were mixed with 100 µL of Folin–Ciocalteu phenol reagent (Sigma Aldrich Co., St. Louis, MO, USA) and kept for 5 min in dark. Then, 80 μL of 7.5% (w/v) Na_2_CO_3_ solution was added and the reaction was further incubated for 30 min at room temperature in dark. The reaction absorbance was measured at 750 nm using an automated microplate reader and the data were expressed as mg of gallic acid equivalents (GAE) per g of extract.

#### 4.7.2. Total Flavonoid Content (TFC)

TFC of the NCC and LpCC extracts was estimated following the protocol adapted from Suh et al. [43]. In a 96-well microtiter plate, the reaction mixture was made by adding 180 μL of 90% diethylene glycol, 20 μL of 1 N NaOH, and 20 μL of sample extracts (50 mg/mL), followed by 60 min incubation at room temperature. The absorbance was recorded at 405 nm using an automated microplate reader. The TFC data were expressed as mg of quercetin equivalents (QE) per g of extract.

#### 4.7.3. Thiol and Allicin Contents

The thiol and allicin contents of the CC juice extracts were measured colorimetrically as described by previously Han et al. [45] and Yang et al. [34]. The thiols were analyzed by adding 100 μL of samples (10 mg/mL) into 100 μL of 1.5 mM 5, 5′-dithios-bis-(2-nitrobenzoic acid) (DTNB) in a 96-well microtiter plate, and the reaction mixture was incubated for 10 min at room temperature. The reaction absorbance was then measured at 412 nm using an automated microplate reader and expressed as micromolar of cysteine equivalents (CE). The total allicin content was determined by estimating the residual cysteine concentration. In brief, a 100 μL aliquot of each sample was mixed with 250 μM of L-cysteine. Following the 10 min incubation, 100 μL of this mixture was added to 100 μL of 1.5 mM DTNB, and the reaction was further incubated for 10 min at room temperature. The absorbance was recorded at 412 nm using an automated microplate reader and the allicin contents were calculated using the Equation (2).
(2)$$Allicin\;contents\;\left(µM\right)=\frac{C-\left(b-a\right)}{2}$$
where, *a*: thiols in samples, *b*: thiols after reaction with L-cysteine, and c: added L-cysteine.

#### 4.7.4. UHPLC-LTQ-Orbitrap-MS/MS Analysis

Metabolite profiling of LpCC and NCC juice extracts (10 mg/mL) was done by using UHPLC-LTQ-Orbitrap-MS/MS. Chloramphenicol (2.5 mg/mL) was used as an internal standard (IS). The sample injection volume of 5 μL was injected at the flow rate of 0.3 mL/min into a UHPLC system equipped with a Vanquish binary pump H system (Thermo Fisher Scientific, Waltham, MA, USA), an auto-sampler, and the column compartment. The chromatographic separation was achieved on a Phenomenex KINETEX^®^ C18 column (100 mm × 2.1 mm, 1.7 μm particle size; Torrance, CA, USA) maintained at the column temperature of 40 °C. The mobile phase consisted of 0.1% formic acid in water (*v/v*, solvent A) and 0.1% formic acid in acetonitrile (*v/v*, solvent B). The mobile phase solvent gradient was programmed as follows: 5% solvent B for 1 min, gradually increased to 100% solvent B over 9 min, and maintained as such for next 1 min followed by a decrease to 5% solvent B over 3 min. The MS data were collected by using an Orbitrap Velos ProTM system equipped with an ion-trap mass spectrometer and HESI-II probe. The mass spectra were acquired over the range of 100–2000 m/z. The probe heater and capillary temperatures were set to 300 °C and 350 °C, respectively. The capillary voltage was set to 2.5 KV and 3.7 KV in negative and positive ion modes, respectively.

#### 4.7.5. Data Processing and Statistical Analyses

UHPLC-LTQ-Orbitrap-MS/MS data were acquired with Xcalibur software (version 2.00, Thermo Fisher Scientific) and converted into netCDF format (∗.cdf) using the Xcalibur software. Peak detection, retention time correction, and alignment were conducted using the MetAlign software package (http://www.metalign.nl). The resulting data matrices were exported to an Excel file format and multivariate statistical analyses including PCA and OPLS-DA were performed using SIMCA-P+12.0 software (Umetrics; Umeå, Sweden). The data sets were autoscaled (univariance scaling) and mean-centered in a column-wise fashion. The quality of the model was evaluated by R2X, R2Y and Q2, which indicate the fitness and prediction accuracy of the model. In addition, the quality of the model was evaluated by *p*-value derived from cross-validation analysis. The significantly discriminant metabolites between experimental groups were selected based on the VIP values, and the significance was tested by one-way analysis of variance (ANOVA) and Student’s t-test using PASW Statistics 18 software (SPSS, Inc., Chicago, IL, USA). The metabolites evaluated by UHPLC-LTQ-Orbitrap-MS/MS were tentatively identified through comparing their MW, RT, and mass fragmentation patterns (MS_n_) data with those obtained from published literature, the in-house library, and available online databases. Further, the elemental compositions and molecular formulae for the candidate metabolites were examined using the Q-Exactive data with low mass errors, i.e., ≤ 5 ppm. The relative abundance of significantly discriminant metabolites was normalized with the IS. The box and whisker plots of metabolite abundance were constructed using STATISTICA 7 software (StatSoft Inc., Tulsa, OK, USA).

All the analytical (TPC, TFC, allicin, and thiol quantification) and bioactivity (antimicrobial, antiviral, and antioxidant) related experiments were performed in three independent replicates. Data were expressed as mean ± SD and statistical significance was assessed using Students’ t-test at *p* < 0.05 level (SPSS 25, IBM, USA).

## 5. Conclusions

To sum up, our findings offer a simple and sustainable strategy to tap and potentiate the application of *Allium tuberosum* (Chinese chives, CC) toward poultry feed application. Controlled *L. plantarum* fermentation of CC juice significantly improved its antibacterial and antiviral efficacy through the synergistic actions of functional metabolites including organic acid, flavonol glycoside, fatty acid, and saponin as revealed by LC-MS/MS analyses. Furthermore, we hypothesize that the fermented CC juice with antipathogenic characteristics may represent a promising and environment-friendly antibiotic alternative for poultry and its applications can also be extended to treat zoonotic diseases.

## Figures and Tables

**Figure 1 antibiotics-09-00386-f001:**
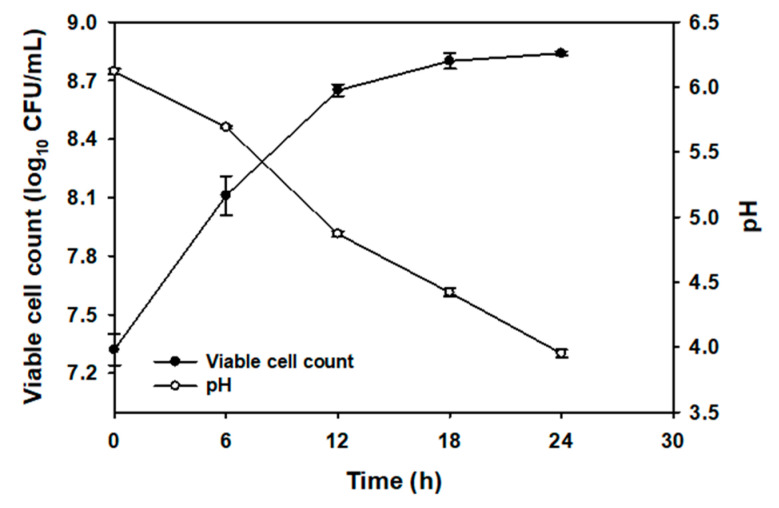
Changes in *L. plantarum* growth and pH during 24 h of fermentation of Chinese chive (CC) juice (25%, volume per volume (*v/v*)).

**Figure 2 antibiotics-09-00386-f002:**
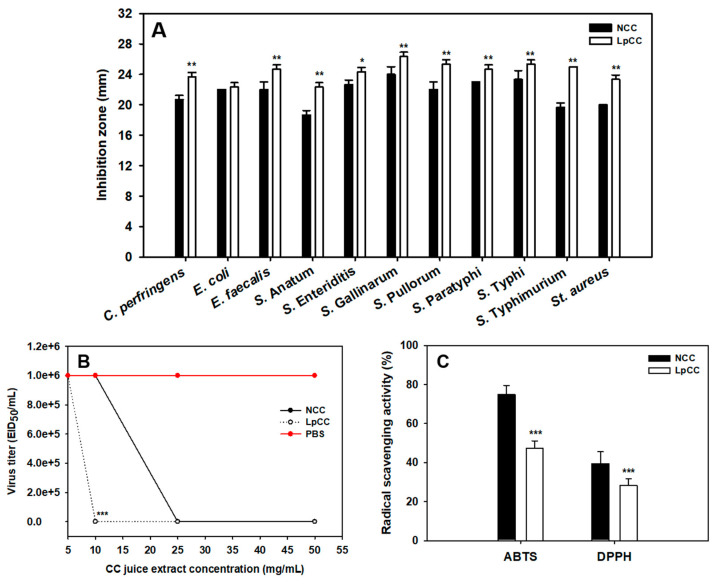
Changes in the bioactivities of CC juice following 24 h of *L. plantarum* fermentation. (**A**) Antibacterial activity of NCC and LpCC supernatants against poultry pathogens using agar well diffusion assay; (**B**) *in ovo* antiviral activity of NCC and LpCC extracts against low-pathogenic avian influenza virus, H1N1 using hemagglutination assay, EID_50_/_mL_: 50% egg infective dose per mL and (**C**) antioxidant activity of NCC and LpCC extracts using 2,2’-azino-bis(3-ethylbenzothiazoline-6-sulfonic acid (ABTS) and 2,2-diphenyl-1-picrylhydrazyl (DPPH) radical scavenging assays. (* *p* < 0.05, ** *p* < 0.01, *** *p* < 0.001), Statistically significant compared with NCC.

**Figure 3 antibiotics-09-00386-f003:**
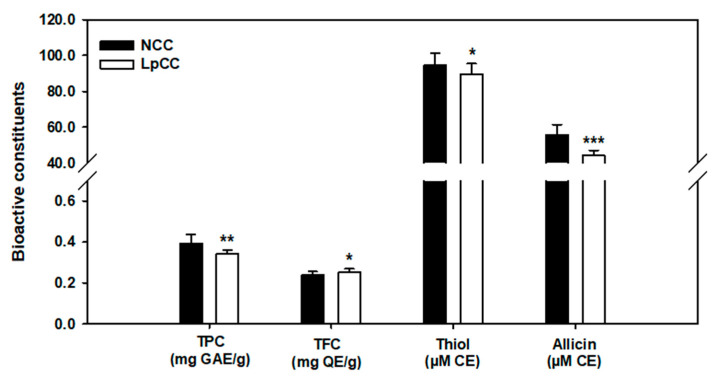
Changes in the biochemical constituents of CC juice following *L. plantarum* fermentation. TPC: total phenolic content expressed in mg of gallic acid equivalent (GAE) per gram of extract; TFC: total flavonoid content expressed in mg of quercetin equivalent (QE) per gram of extract; thiol and allicin are expressed as micromolar of cysteine equivalent (CE). (* *p* < 0.05, ** *p* < 0.01, *** *p* < 0.001), Statistically significant compared with the NCC.

**Figure 4 antibiotics-09-00386-f004:**
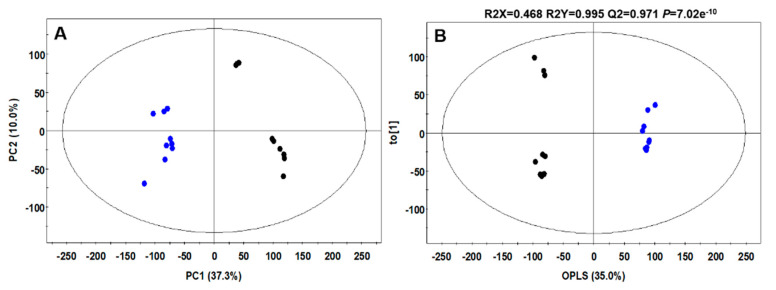
(**A**) Principal component analysis (PCA) and (**B**) orthogonal partial least squares-discriminant analysis (OPLS-DA) score plots derived from ultra-high-performance liquid chromatography-linear trap quadrupole-orbitrap-tandem mass spectrometry (UHPLC-LTQ-Orbitrap-MS/MS) datasets of NCC and LpCC extracts in negative ion mode (●,NCC; ●, LPCC).

**Figure 5 antibiotics-09-00386-f005:**
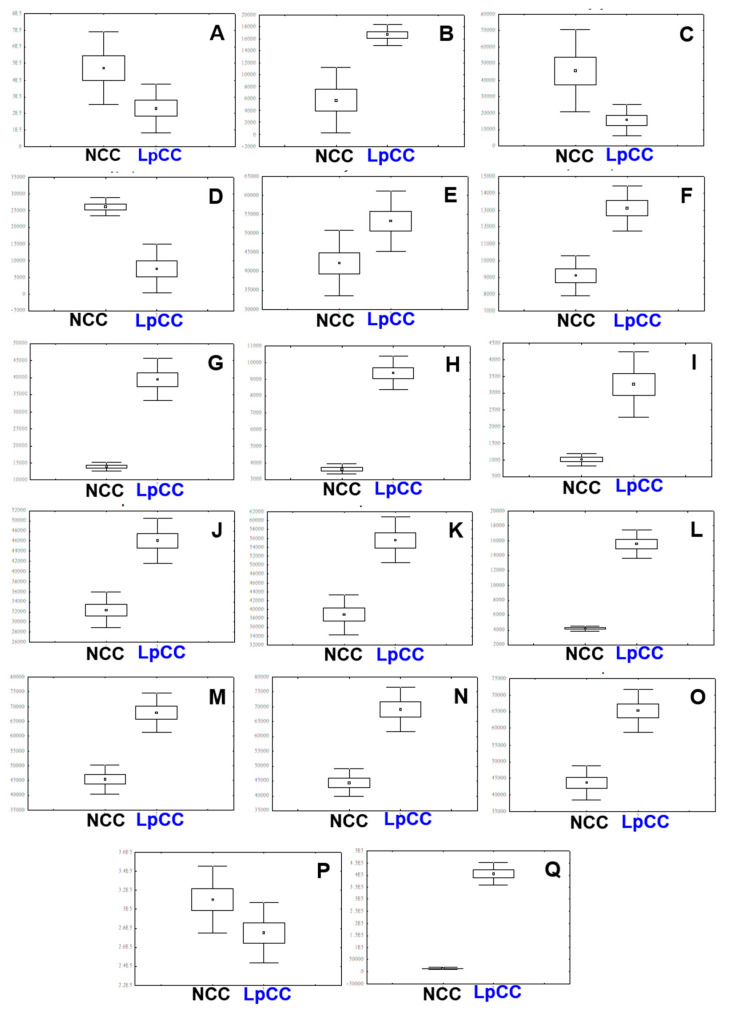
Box-and-whisker plots illustrating the relative abundance of metabolite levels (VIP > 0.7, *p* < 0.05) of NCC and LpCC extracts using UHPLC-LTQ-Orbitrap-MS/MS. (**A**) N-(1-deoxy-1-fructosyl)phenylalanine, (**B**) 3-(2,3,4-trihydroxy-5-methoxyphenyl)propanoic acid, (**C**) glycyrol, (**D**) tryptophan, (**E**) benzoylmesaconine derivative, (**F**) kaempferol-diglucoside, (**G**) isorhamnetin 3,4’-diglucoside, (**H**) quercetin-diglucoside, (**I**) quercetin-hexoside, (**J**) saponin 1, (**K**) saponin 2, (**L**) kaempferol-glucoside, (**M**) saponin 3, (**N**) saponin 4, (**O**) saponin 5, (**P**) tianshic acid, and (**Q**) 12-hydroxystearic acid.

**Table 1 antibiotics-09-00386-t001:** Antimicrobial activity of the isolated microorganisms against poultry pathogenic bacteria.

Isolates	Poultry Pathogens										
	*Salmonella* Gallinarum	*Salmonella* Enteritidis	*Salmonella* Pullorum	*Salmonella* Typhimurium	*Salmonella* Anatum	*Salmonella* Typhi	*Salmonella* Paratyphi	*Clostridium perfringens*	*Staphylococcus aureus*	*Escherichia coli*	*Enterococcus faecalis*
*Leuconostoc mesenteroides* SK4645	-	+	-	-	-	-	+	-	-	+	-
*Lactobacillus sakei* SK4688	-	+	-	+	-	-	-	-	-	-	-
*Lactobacillus plantarum* SK4719	+	+	+	+	+	+	+	+	+	+	+
*Weissella cibaria* SK4720	+	-	-	-	+	-	-	-	-	-	-
*Weissella paramesenteroides* SK4721	+	-	-	-	+	-	-	-	-	+	-
*Bacillus megaterium* SK4723	-	-	+	-	-	+	-	-	+	-	-
*Bacillus aryabhattai* SK4724	-	-	-	+	-	-	-	-	-	-	-
*Bacillus pumilus* SK4726	-	+	-	-	-	-	-	+	-	-	-
*Bacillus subtilis* SK4730	-	-	-	-	-	-	-	-	-	-	-
*Staphylococcus sciuri* SK4727	-	-	-	-	-	-	-	+	-	-	-
*Micrococcus luteus* SK4728	-	-	-	-	-	-	-	-	-	-	-
*Saccharomyces cerevisiae* SK4690	-	-	-	-	-	+	-	-	+	-	-

-: no activity; +: activity.

**Table 2 antibiotics-09-00386-t002:** Tentatively identified metabolites in CC juice extracts based on the UHPLC-LTQ-Orbitrap-MS/MS analyses.

No.	Tentatively Identified Metabolites	RT (min)	MW	Measured Mass	MS/MS Fragments	Class of Compounds
				Negative Mode (*m/z*) *		
1	N-(1-Deoxy-1-fructosyl)phenylalanine	1.09	327	326.1204	326 > 308/278/236/206/164	Amino acid
2	3-(2,3,4-trihydroxy-5-methoxyphenyl)propanoic acid	1.75	228	227.1379	227 > 183/209	Organic acid
3	Glycyrol	1.83	366	365.1305	365 > 275/347/203/317	Coumestan
4	Tryptophan	1.92	204	203.0811	203 > 159/116/142/186	Amino acid
5	Benzoylmesaconine derivative	3.79	559	558.2698	558 > 540/514/496/470/452/395	Alkaloid
6	Feruloyl-galactaric acid	3.97	386	385.0720	385 > 191/209/367	Organic acid
7	Kaempferol-diglucoside	4.00	610	609.1408	609 > 447/285/489/581	Flavonol
8	Isorhamnetin 3,4’-diglucoside	4.21	640	639.3306	639 > 621/579/549/519/477	Flavonol
9	Quercetin-diglucoside	4.33	626	625.1360	625 > 463/300/445/505/607	Flavonol
10	Quercetin-hexoside	4.76	464	463.0843	463 > 301	Flavonol
11	Kaempferol diglucoside-(feruloylglucoside)	4.77	948	947.2372	94 7> 623/785/447/609/285	Flavonol
12	Saponin 1	4.92	808	807.4156	807 > 789/763/717/645	Saponin
13	Saponin 2	4.99	852	851.4416	851 > 833/807/761/689/512	Saponin
14	Kaempferol-glucoside	5.01	448	447.0897	447 > 284/255	Flavonol
15	Saponin 3	5.05	896	895.4661	895 > 877/859/763/745	Saponin
16	Saponin 4	5.11	940	939.4932	939 > 921/895/848/776	Saponin
17	Saponin 5	5.17	984	983.5189	983 > 789/803/771/951/821	Saponin
18	Oxo-dihydroxy-octadecenoic acid	6.25	328	327.2145	327 > 309/291/273/247/239	Fatty acid
19	Tianshic acid	6.50	330	329.2291	329 > 311/293/229/211/171	Fatty acid
20	12-Hydroxystearic acid	9.66	300	299.2562	299 > 281/253	Fatty acid

* [M-H]^-^: Ion detected is one unit lower than the monoisotopic mass of the uncharged molecule; RT: retention time; MW: molecular weight.

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
