# Peer review of "Controlled Fermentation Using Autochthonous Lactobacillus plantarum Improves Antimicrobial Potential of Chinese Chives against Poultry Pathogens"

_antibiotics, 2020, doi:10.3390/antibiotics9070386_

Round 1
Reviewer 1 Report
The study of Damini Kothari and colleagues, describe the differences between the non-fermented Chinese chives (NCC) and the fermented one with the autochthonous Lactobacillus plantarum (LpCC). Interestingly, the authors demonstrate that upon the fermentation the antimicrobial activity of CC extract increase, otherwise the antioxidant activity decrease significantly. Furthermore, by using mass spectrometry based untargeted metabolomic profile, the differences between the extracts obtained from NCC and LpCC were highlighted. The experiments were well thought out and all the assays were well designed. The paper can not be accepted in the current form and a minor revision is needed.
Major comments
- In the introduction section the authors claim that, spontaneous and uncontrolled fermentation could have microbiological and toxicological side effects. I have questioned why the authors should not test the possible toxic effects on eukaryotic cell lines, even better if derived from poultry, of the LpCC extract? I think that the authors could at least discuss on that.
Additional points
- The figure numbering should be revised because the first three figures are all numbered as Figure 1
- Line 363 “we” should be revised in “were”
Author Response
Answers to the reviewer's comments (Manuscript ID: antibiotics-851561)
We are extremely thankful to the concerned reviewer for considering our manuscript toward revision. We are grateful for all the queries and suggestions by the reviewer that helped us to improve the original version of the manuscript. We hope that all queries, comments, and suggestions were adequately justified in the revised version of the manuscript. Moreover, we also self-assessed our manuscript again and incorporated some corrections in its revised version to increase the readability of the manuscript.
NOTE: All the changes made in the revised manuscript are track-changed.
The study of Damini Kothari and colleagues, describes the differences between the non-fermented Chinese chives (NCC) and the fermented one with the autochthonous Lactobacillus plantarum (LpCC). Interestingly, the authors demonstrate that upon the fermentation the antimicrobial activity of CC extract increase, otherwise the antioxidant activity decrease significantly. Furthermore, by using mass spectrometry based untargeted metabolomic profile, the differences between the extracts obtained from NCC and LpCC were highlighted. The experiments were well thought out and all the assays were well designed. The paper cannot be accepted in the current form and a minor revision is needed.
Major comments
In the introduction section the authors claim that, spontaneous and uncontrolled fermentation could have microbiological and toxicological side effects. I have questioned why the authors should not test the possible toxic effects on eukaryotic cell lines, even better if derived from poultry, of the LpCC extract? I think that the authors could at least discuss on that.
Response: In this study, the main objective was to select an autochthonous culture exhibiting antimicrobial activity and its subsequent application to ferment Chinese chives with improved antimicrobial potential. Therefore, this aspect was not included.
Moreover, we checked the antiviral activity of CC juice extracts in chick embryos and therein we did not find any toxic effects using the concentration of the extracts up to 50 mg/ml (1 to 10 mg/chick embryo).
In the revised manuscript, we included some discussion on the microbiological and toxicological side effects of spontaneous fermentation.
“Spontaneous fermentation may not be an authentic system to obtain a safe and palatable final product as the possibility of detrimental microbiological/toxicological consequences such as biogenic amines, mycotoxins, and pathogenic microbes, could not be excluded. Recently, Lee et al. (2019) indicated the involvement of autochthonous LAB in biogenic amine accumulation during spontaneous fermentation of green onion kimchi.”
Additional points
The figure numbering should be revised because the first three figures are all numbered as Figure 1
Response: We are sorry for this mistake and numbering of the figures has been corrected in the revised manuscript.
Line 363 “we” should be revised in “were”
Response: We are sorry for this mistake and has been corrected in the revised manuscript (Line 394).
We welcome the detailed and constructive comments from the reviewer, and we have attempted our best to justify those in the revised manuscript addressing most of the concerns. However, we welcome any further queries and constructive suggestions posed by the reviewer.
Thank you for your consideration. I look forward to hearing from you.
Yours Sincerely,
Soo-Ki Kim, Ph.D. Professor
Department of Animal Science and Technology
Konkuk University, Seoul 05029, Korea
E-mail: sookikim@konkuk.ac.kr
Tel: +82-2-450-3728; Fax: +82-2-458-3728

Reviewer 2 Report
The work investigates and characterizes the antimicrobial effect and antioxidant activity of fermented Chinese chives extract by the autochthonous Lactobacillus plantarum. This manuscript addresses an area of interest, it is clearly written and I have enjoyed reading it. I have some minor comments that could improve the study.
Abstract: please describe LpCC, LC-MS, PC, PLS-DA, NCC
Line 54: plant genus
Line 59: I would change trivially for known as
Line 63: has instead of have: the fermentation has enhanced…
Line 68 and 247: why the fermentation being palatable is important for poultry? Would they not eat it if it weren’t?
Line 91: strains
Line 93: and was hence selected
Lines 104, 105 and Figure 1: the numbers seem to be incorrect, I would say either:
-decreased to 5.7 from an initial value of 6.1 within 6h of fermentation
-decreased to 4.9 from an initial value of 6.1 within 12h of fermentation
Line 115 and Figure 1A: S. aureus, not St. Also, write the full name on first appereance, Clostridium perfringens, enterococcus faecalis…
Line 110 and throughout the manuscript: the material and methods section seem to have been moved from after the introduction to after the discussion, as a consequence all the acronyms are explained after they appear on the text. Please explain them on their first appearance. i.e:
- Line 110: describe NCC and LpCC
- Line 135: describe ABTS and DPPH
- Line 147: describe UHPLC-LTQ-Orbitrap MS/MS
- Line 148: describe PCA
- Line 150: describe OPLS-DA
- Line 165: No need to describe here LpCC and NCC, they have already appeared on the text
- Line 166: describe VIP
- Explain CFU
- Line 281 no need to describe here CC
- Line 304 no need to describe here NCC
- Line 308 no need to describe here LpCC
- Line 328: describe EID
- Line 331 no need to describe here DPPH and ABTS
- Line 348: no need to describe LpCC, NCC here
- Line 374: no need to describe UHPLC-LTQ-Orbitrap MS/MS here
- Line 401: no need to describe VIP here
The numbering of the figures is incorrect, Figure 2 and 3 appear as figure 1.
Line 121, 122: there is no need to explain LpCC and NCC so many times in this figure. They are already described in line 119 and that is enough. Figure 2C Describe ABTS and DPPH.
Line 136 compared to what? NCC?
Figure 3: describe TPC, TFC, GAE, CE
Line 154-155: according to Table 2 there are 6 flavonoids, not 7 and 2 organic acids, not 1.
Figure 4, Line 150: please, decide if you want to use OPLS-DA or PLS-DA. Also, what is figure 4A, you only explain figure 4B.
Table 2: what is M-H?
Lines 164-167, 181 and 412: significantly different?
Lines 169-174: what are the numbers in bold and between parentheses? Graphs in figure 5? I would use letters to be consistent with other figures and prevent confusion with the references. Use 5A, 5B, 5C… also, make sure to add those letters in the actual figure, Figure 5 graphs are unlabeled currently.
Figure 6: why is Pearson’s correlation value classified from -1.5 to 1.5 if 1 is the maximum value? It is my understanding that it should go from 1 to -1.
Also, please, in order to differentiate the saponins, could you label them as saponin 1, saponin 2…
Lines 207-209, 245, 259, 262 Put in vitro, in vivo, Sophora japonica, in ovo… in cursive
Line 202, 216 log10 CFU/mL
Line 222 decline quickly and are even lost
Line 246 In line with this consideration
Line 293 and Table 1: faecalis
Line 308: why were not the samples filtered after fermentation? Even at such high g bacteria can still appear in the supernatant and interfere with the results.
Line 323 were instead of are
Line 236 from olives
Line 271 have
Author Response
Answers to the reviewer's comments (Manuscript ID: antibiotics-851561)
We are extremely thankful to the concerned reviewer for considering our manuscript toward revision. We are grateful for all the queries and suggestions by the reviewer that helped us to improve the original version of the manuscript. We hope that all queries, comments, and suggestions were adequately justified in the revised version of the manuscript. Moreover, we also self-assessed our manuscript again and incorporated some corrections in its revised version to increase the readability of the manuscript.
NOTE: All the changes made in the revised manuscript are track-changed.
The work investigates and characterizes the antimicrobial effect and antioxidant activity of fermented Chinese chives extract by the autochthonous Lactobacillus plantarum. This manuscript addresses an area of interest, it is clearly written, and I have enjoyed reading it. I have some minor comments that could improve the study.
Response: Thank you for your kind appraisal of our work.
Abstract: please describe LpCC, LC-MS, PLS-DA, NCC
Response: We have provided the description of all the abbreviations in the abstract of revised manuscript.
Line 54: plant genus
Response: The suggested change has been included in the revised manuscript (Line 63).
Line 59: I would change trivially for known as
Response: The suggested change has been included in the revised manuscript (Line 68).
Line 63: has instead of have: the fermentation has enhanced…
Response: The suggested change has been included in the revised manuscript (Line 73).
Line 68 and 247: why the fermentation being palatable is important for poultry? Would they not eat it if it weren’t?
Response: Fermentation is an olden technology to improve the sensory and functional properties of food products. In this study, we tried to improve the antimicrobial properties of CC following fermentation using a starter exhibiting antimicrobial activity against avian pathogens.
Line 68: The uncontrolled/spontaneous fermentation results in higher concentrations of both acetic acid and biogenic amines which adversely affect the palatability of fermented liquid feed diets.
Line 127: Generally, Alliums are rich in organosulfur compounds (OSC), which is responsible for their strong odor, which is reported to affect the feed intake in broiler if used as fresh in high concentrations. The L. plantarum fermentation could reduce the levels these OSCs, thereby might improve the feed palatability.
Line 91: strains
Response: The suggested change has been included in the revised manuscript (Line 103).
Line 93: and was hence selected
Response: The suggested change has been included in the revised manuscript (Line 106).
Lines 104, 105 and Figure 1: the numbers seem to be incorrect, I would say either:
-decreased to 5.7 from an initial value of 6.1 within 6h of fermentation
-decreased to 4.9 from an initial value of 6.1 within 12h of fermentation
Response: The line has been corrected in the revised manuscript as suggested by the reviewer (Line 117).
Line 115 and Figure 1A: S. aureus, not St. Also, write the full name on first appearance, Clostridium perfringens, enterococcus faecalis…
Response: Yes, we do agree with the review that it should be S. aureus not St. aureus, but to differentiate from Salmonella (S.) we have written as St. for Staphylococcus.
The microorganism names were written in full on their first appearance in the revised manuscript (Table 1).
Line 110 and throughout the manuscript: the material and methods section seem to have been moved from after the introduction to after the discussion, as a consequence all the acronyms are explained after they appear on the text. Please explain them on their first appearance. i.e:
Response: Yes, this has been done by mistake and hence corrected throughout the manuscript.
- Line 110: describe NCC and LpCC
Response: NCC and LpCC are described as non-fermented CC juice (NCC) and 24 h-L. plantarum fermented CC juice (LpCC) in the revised manuscript (Line 123-124).
- Line 135: describe ABTS and DPPH
Response: ABTS and DPPH are described in the revised manuscript (Line 151-152).
- Line 147: describe UHPLC-LTQ-Orbitrap MS/MS
Response: UHPLC-LTQ-Orbitrap MS/MS is described in the revised manuscript (Line 166-167).
- Line 148: describe PCA
Response: PCA is described in the revised manuscript (Line 168).
- Line 150: describe OPLS-DA
Response: OPLS-DA is described in the revised manuscript (Line 171).
- Line 165: No need to describe here LpCC and NCC, they have already appeared on the text.
Response: The suggested change has been included in the revised manuscript.
- Line 166: describe VIP
Response: VIP is described in the revised manuscript (Line 189).
- Explain CFU
Response: CFU is described in the revised manuscript (Line 114).
- Line 281 no need to describe here CC
Response: The suggested change has been included in the revised manuscript.
- Line 304 no need to describe here NCC
Response: The suggested change has been included in the revised manuscript.
- Line 308 no need to describe here LpCC
Response: The suggested change has been included in the revised manuscript.
- Line 328: describe EID
Response: EID50 indicates egg infective dose 50% per mL. It is explained in the legend of Figure 2C in the revised manuscript (Line 135).
- Line 331 no need to describe here DPPH and ABTS
- Line 348: no need to describe LpCC, NCC here
Response: The suggested change has been included in the revised manuscript.
- Line 374: no need to describe UHPLC-LTQ-Orbitrap MS/MS here
Response: The suggested change has been included in the revised manuscript.
- Line 401: no need to describe VIP here
Response: The suggested change has been included in the revised manuscript.
The numbering of the figures is incorrect, Figure 2 and 3 appear as figure 1.
Response: This was done by mistake; we have now corrected the numbering of figures in the revised manuscript.
Line 121, 122: there is no need to explain LpCC and NCC so many times in this figure. They are already described in line 119 and that is enough. Figure 2C Describe ABTS and DPPH.
Response: Ok. We have removed the repeated explanation for LpCC and NCC from the revised manuscript. We have included the description of ABTS and DPPH in the figure legend in the revised manuscript (Line 137-138).
Line 136 compared to what? NCC?
Response: Yes. We have included now “compared to NCC” (Line 152).
Figure 3: describe TPC, TFC, GAE, CE
Response: TPC, TFC, GAE, and CE have been described in the legend of Figure 3 of revised manuscript as “TPC: total phenolic content expressed in mg of gallic acid equivalent (GAE) per gram of extract; TFC: total flavonoid content expressed in mg of quercetin equivalent (QE) per gram of extract; thiol and allicin are expressed as micromolar of cysteine equivalent (CE)” (Line 160-163).
Line 154-155: according to Table 2 there are 6 flavonoids, not 7 and 2 organic acids, not 1.
Response: This was be done by mistake and now has been corrected in the revised manuscript (Line 175, 176).
Figure 4, Line 150: please, decide if you want to use OPLS-DA or PLS-DA. Also, what is figure 4A, you only explain figure 4B.
Response: We used now OPLS-DA throughout the manuscript. We also now explained the Figure 4A in the revised manuscript.
Table 2: what is M-H?
Response: M-H indicates ion detected is one unit lower than the monoisotopic mass of the uncharged molecule and it has been included in the revised manuscript (Line 185).
Lines 164-167, 181 and 412: significantly different?
Response: In Lines 164-167, No, it is significantly discriminant. The multivariate analysis, a discriminant function analysis (PCA and OPLS-DA) is used to identify number of variables discriminate between two or more groups. Herein, the multivariate analysis (PCA and OPLS-DA), that we have performed were statistically significant. First, we identified the significant variables i.e. metabolites, then we checked the relative levels of metabolites were significantly different or not between NCC and LpCC extracts. Therefore, the metabolites selected were significantly discriminant at VIP>0.7 and p<0.05 and the relative levels were significantly different (p<0.05).
Lines 169-174: what are the numbers in bold and between parentheses? Graphs in figure 5? I would use letters to be consistent with other figures and prevent confusion with the references. Use 5A, 5B, 5C… also, make sure to add those letters in the actual figure, Figure 5 graphs are unlabeled currently.
Response: The numbers indicate the serial no. of compounds identified from LC-MS/MS analysis in Table 2.
The labelling of the Figure 5 has been done as Figure 5A-Q according to the reviewer comment (Line 201-205).
Figure 6: why is Pearson’s correlation value classified from -1.5 to 1.5 if 1 is the maximum value? It is my understanding that it should go from 1 to -1.
Response: As per the other reviewer comments, the Figure has been removed from the manuscript.
Also, please, in order to differentiate the saponins, could you label them as saponin 1, saponin 2…
Response: The suggested change has been included in the revised manuscript (Table 2).
Lines 207-209, 245, 259, 262 Put in vitro, in vivo, Sophora japonica, in ovo… in cursive
Response: The suggested changed have been incorporated in the revised manuscript (Line 235, 237, 287, and 290).
Line 202, 216 log10 CFU/mL
Response: In Line 230 of revised manuscript, it is log10 CFU/g and while in line 240, the suggested change is being done (Line 230, 244).
Line 222 decline quickly and are even lost
Response: The suggested change has been included in the revised manuscript (Line 250).
Line 246 In line with this consideration
Response: The suggested change has been included in the revised manuscript (Line 274).
Line 293 and Table 1: faecalis
Response: The suggested change has been included in the revised manuscript (Line 322).
Line 308: why were not the samples filtered after fermentation? Even at such high g bacteria can still appear in the supernatant and interfere with the results.
Response: The supernatants were filtered with 0.22 membrane filters and this is now included in the revised manuscript (Line 338).
Line 323 were instead of are
Response: The suggested change has been included in the revised manuscript (Line 352).
Line 236 from olives
Response: The suggested change has been included in the revised manuscript (Line 268).
Line 271 have
Response: The suggested change has been included in the revised manuscript (Line 299).
We welcome the detailed and constructive comments from the reviewer, and we have attempted our best to justify those in the revised manuscript addressing most of the concerns. However, we welcome any further queries and constructive suggestions posed by the reviewer.
Thank you for your consideration. I look forward to hearing from you.
Yours Sincerely,
Soo-Ki Kim, Ph.D. Professor
Department of Animal Science and Technology
Konkuk University, Seoul 05029, Korea
E-mail: sookikim@konkuk.ac.kr
Tel: +82-2-450-3728; Fax: +82-2-458-3728

Reviewer 3 Report
The title proposed by the authors of "Controlled fermentation of Chinese chives by autochthonous Lactobacillus plantarum toward developing a non-antibiotic feed additive" does not quite coincide with the content of the work. The title has overinterpretation. I suggest a shorter title:
“Controlled fermentation of Chinese chives by autochthonous Lactobacillus plantarum”
Line 58: Salmonella Typimurium. “Typhimurium” is the name of a serotype, not a species. The names of serotype are written in capital letters without the italic font.
Line: Tabela 1.
The names of the microorganisms are incorrect. It should be S. Enteritidis (Enteritidis is a serotype, not a species). The same applies to S. Typhi, S. Paratyphi, and Typhimuroim.
In the current nomenclature, we distinguish only two species of Salmonella: Salmonella enterica and Salmonella bongori. Salmonella strains are differentiated within these two species into subspecies, e.g. S. enterica subsp. salmae or S.enterica Typhimurium serotype.
Please sort the nomenclature of microorganisms.
Antagonistic activity should be shown as growth inhibition zone and standard deviation. Please enter specific values in the table.
Line: Figure 6: The diagram is illegible and can be successfully removed from work. Correlations should be described by mathematical models. These are extremely difficult issues with many variables. There are many variables here and I have doubts about these correlations. I suggest removing this item from work.
Line 209: „in vitro” italic font
Line: 216-217: “which can prevent proliferation of spoilage organisms and thereby maintaining the quality characteristics related to yeast outgrowth during the 218 storage period” – there is no justification in the results.
Line: 223-224 Please avoid referring to unpublished data. It is difficult to verify. I recommend removing such fragments.
Line: 225-229: “The stronger and persistent antibacterial activity of LpCC might be attributed to lactic acid and short-chain organic acids, released following the fermentation. In addition, the antimicrobial activities of the fermented CC juice could also be attributed to the specific compositional changes in CC juice following the L. plantarum mediated fermentation. “
The Authors have not studied the acid metabolite profile, therefore they cannot rely on such data. Low pH is not enough for the above.
Line: 291-292 Please correct the names of the microorganisms in accordance with the remarks given above.
After applying the indicated corrections, I recommend accepting the manuscript. Currently minor revision.
Author Response
Answers to the reviewer's comments (Manuscript ID: antibiotics-851561)
We are extremely thankful to the concerned reviewer for considering our manuscript toward revision. We are grateful for all the queries and suggestions by the reviewer that helped us to improve the original version of the manuscript. We hope that all queries, comments, and suggestions were adequately justified in the revised version of the manuscript. Moreover, we also self-assessed our manuscript again and incorporated some corrections in its revised version to increase the readability of the manuscript.
NOTE: All the changes made in the revised manuscript are track-changed.
The title proposed by the authors of "Controlled fermentation of Chinese chives by autochthonous Lactobacillus plantarum toward developing a non-antibiotic feed additive" does not quite coincide with the content of the work. The title has overinterpretation. I suggest a shorter title:
“Controlled fermentation of Chinese chives by autochthonous Lactobacillus plantarum”
Response: We have changed the title with more precise outcome of our study as
“Controlled fermentation using autochthonous Lactobacillus plantarum improves antimicrobial potential of Chinese chives against poultry pathogens”
Line 58: Salmonella Typimurium. “Typhimurium” is the name of a serotype, not a species. The names of serotype are written in capital letters without the italic font.
Line: Table 1.
The names of the microorganisms are incorrect. It should be S. Enteritidis (Enteritidis is a serotype, not a species). The same applies to S. Typhi, S. Paratyphi, and Typhimuroim.
In the current nomenclature, we distinguish only two species of Salmonella: Salmonella enterica and Salmonella bongori. Salmonella strains are differentiated within these two species into subspecies, e.g. S. enterica subsp. salmae or S.enterica Typhimurium serotype.
Please sort the nomenclature of microorganisms.
Response: We have corrected the names of the pathogens as suggested by the reviewer (Table 1).
Antagonistic activity should be shown as growth inhibition zone and standard deviation. Please enter specific values in the table.
Response: In Table 1, our purpose was to screen the isolates based on presence or absence of antimicrobial activity and therefore zones of inhibition were not recorded. Therefore, we did not have the data as asked by the reviewer.
Line: Figure 6: The diagram is illegible and can be successfully removed from work. Correlations should be described by mathematical models. These are extremely difficult issues with many variables. There are many variables here and I have doubts about these correlations. I suggest removing this item from work.
Response: The figure 6 along with its method, results and discussion parts have been successfully removed from the manuscript as suggested by the reviewer.
Line 209: “in vitro” italic font
Response: It has been done throughout in the revised manuscript.
Line: 216-217: “which can prevent proliferation of spoilage organisms and thereby maintaining the quality characteristics related to yeast outgrowth during the storage period” – there is no justification in the results.
Response: We are trying to say that this may be possible, in the revised manuscript the word “can” is being replaced by “may”.
Line: 223-224 Please avoid referring to unpublished data. It is difficult to verify. I recommend removing such fragments.
Response: This sentence has been deleted in the revised manuscript.
Line: 225-229: “The stronger and persistent antibacterial activity of LpCC might be attributed to lactic acid and short-chain organic acids, released following the fermentation. In addition, the antimicrobial activities of the fermented CC juice could also be attributed to the specific compositional changes in CC juice following the L. plantarum mediated fermentation. “
The Authors have not studied the acid metabolite profile, therefore they cannot rely on such data. Low pH is not enough for the above.
Response: We do agree low pH alone is not sufficient, but we did measure the titratable acidity (TA), which is an indicator of total acidity [TA (g/100ml): NCC, 0.336 ± 0.01; LpCC, 1.767 ± 0.01]. Based on this, we hypothesized our results, if the reviewer wants us to include the data of the titratable acidity, we will include the same.
Line: 291-292 Please correct the names of the microorganisms in accordance with the remarks given above.
Response: The names of the microorganisms has been corrected as per the above comments.
After applying the indicated corrections, I recommend accepting the manuscript. Currently minor revision.
Response: All the suggested corrections have been implemented in the revised manuscript.
We welcome the detailed and constructive comments from the reviewer, and we have attempted our best to justify those in the revised manuscript addressing most of the concerns. However, we welcome any further query and constructive suggestions posed by the reviewer.
Thank you for your consideration. I look forward to hearing from you.
Yours Sincerely,
Soo-Ki Kim, Ph.D. Professor
Department of Animal Science and Technology
Konkuk University, Seoul 05029, Korea
E-mail: sookikim@konkuk.ac.kr
Tel: +82-2-450-3728; Fax: +82-2-458-3728
